# Insomnia—A Systematic Review and Comparison of Medical Resident’s Average Off-Call Sleep Times

**DOI:** 10.3390/ijerph20054180

**Published:** 2023-02-26

**Authors:** Eve Ardizzone, Emily Lerchbaumer, Johannes C. Heinzel, Natalie Winter, Cosima Prahm, Jonas Kolbenschlag, Adrien Daigeler, Henrik Lauer

**Affiliations:** 1Department of Hand-, Plastic, Reconstructive and Burn Surgery, BG Unfallklinik Tuebingen, University of Tuebingen, Schnarrenbergstraße 95, 72076 Tuebingen, Germany; 2Department of Neurology, Hertie Institute for Clinical Brain Research (HIH), University of Tuebingen, Hoppe-Seyler-Str. 3, 72076 Tuebingen, Germany

**Keywords:** sleep, sleep deprivation, systematic review, residency, sleep times

## Abstract

Sleep deprivation is known to have serious consequences, including a decrease in performance, attention and neurocognitive function. It seems common knowledge that medical residents are routinely sleep deprived, yet there is little objective research recording their average sleep times. To discern whether residents may be suffering from the abovementioned side effects, this review aimed to analyze their average sleep times. Thirty papers recording the average sleep time of medical residents were found via a literature search using the key words “resident” and “sleep”. An analysis of the mean sleep times cited therein revealed a range of sleep from 4.2 to 8.6 h per night, the median being 6.2 h. A sub-analysis of papers from the USA showed barely any significant differences in sleep time between the specialties, but the mean sleep times were below 7 h. The only significant difference (*p* = 0.039) was between the mean sleep times of pediatric and urology residents, with the former achieving less sleep. The comparison of methods for data collection showed no significant difference in the sleep times collected. The results of this analysis imply that residents are regularly sleep deprived and may therefore suffer from the abovementioned consequences.

## 1. Introduction

The American Academy of Sleep Medicine and Sleep Research Society suggests at least 7 h of sleep per night [1] as a healthy amount for most adults. Any less is considered sleep deprivation, which may have various consequences tied to it. While almost 70% of American citizens manage to obtain an adequate amount of sleep at night [2], it has long been a stereotype that sleep deprivation is a side effect of being a medical professional. Residents in particular seem to be hit hardest by this chronic sleep deprivation, which not only affects their ability to care for their patients but can also afflict their personal health and mental wellbeing [3,4]. Among other effects, chronic sleep deprivation may cause lapses in attention, reduced cognitive performance and a worsened working memory [3], thus putting patients at risk of being subjects to mistakes. One study even showed that concentration, alertness and visuo-motor skills after a night shift had decreased to a similar level as those of doctors who performed with a blood alcohol content of 0.6 permille [5], and while a doctor that is intoxicated would never be allowed to treat a patient, sleep deprived doctors routinely do [3,5].

Notably, this comparison of the decrease in performance may only be applicable when comparing the execution of relatively simple tasks. Once more difficult tasks are compared, tired residents make significantly less mistakes than drunk ones, although they still performed worse than their well-rested colleagues [6]. 

Crucially, the available literature evaluating surgical performance when sleep deprived is very conflicting. Many studies show a decrease in performance when surgeons are asked to perform various simulated surgical tasks post call [6,7,8,9,10,11,12,13,14,15,16]. A systematic review of the effects of acute sleep deprivation on surgeons found that of the 21 papers analyzed, 12 showed a negative effect on surgical skills [17]. However, another nine studies analyzed in the same review showed no significant increase in errors [17], a finding backed up by a further review from 2015 which found that while tool movement may be negatively affected by lack of sleep, the overall data evaluating surgical performance were conflicting [18]. One study even found that surgical residents post call performed better than when they were well rested, suggesting that acute sleep deprivation may lead to increased alertness [19].

A further factor that seems to predict whether lack of sleep will affect a resident’s performance is experience. A systematic review analyzing the effects of fatigue on surgical performance and outcomes found that residents with less training or experience appeared to be more affected than their more senior counterparts [20].

On top of this, the number of consecutive nights on call may also influence performance. A 2008 study analyzing the deterioration of performance in newly acquired surgical skills over several successive night shifts showed a significant increase in task completion time and error rates after the first night shift. In subsequent nights, the error rates decreased again and performance shifted back towards baseline levels [13]. One reason as to why surgical performance may not always suffer after a night of sleep deprivation is given by a 2010 neuroimaging study simulating a night shift for surgical residents, suggesting that maintaining the same levels of cognitive performance in a sleep deprived state requires greater attention and concentration compared to performing the same task in a rested state [21]. 

The same study found that well-trained technical tasks elicited less of a response in the prefrontal cortex, implying that they generally require less concentration. As a result, these kinds of routine, manual tasks may not suffer as much in a sleep deprived state [21]. This corresponds to the findings of Tomasko et al. [22], who showed that sleep deprived surgeons, while still being able to complete the tasks given to them without mistakes reported a higher level of subjective mental workload.

It is, however, important to note that many of these studies only take into account the performance of residents immediately after an on-call shift versus after 24 h of rest. Results showing no decrease in skill after acute sleep deprivation merely imply that working a single on-call shift poses no greater risks to patient. These findings cannot necessarily be transferred to the performance of surgeons suffering from chronic sleep deprivation.

As the authors of this review are predominantly surgeons, there is a natural tendency to focus on the negative effects of sleep deprivation on this particular field of medicine. Furthermore, through the simulation trainings mentioned above, a surgeon’s quality of work is more easily measured than that of an internal medicine resident, for instance. Of course, it must be acknowledged that sleep deprivation affects residents of all specialties equally, and a decline in performance for those residents of non-surgical specialties, while not as easily objectified, cannot be discounted.

Further, sleep deprivation also leads to a significant decrease in driving skills and an increase in motor vehicle collisions [16,23,24], putting doctors returning home from their shifts at serious risk of injury or death. As one study of emergency care providers found, 34% of their subjects admitted to falling asleep at the wheel at least once over the past 3 months, and 39% could not recall at least one of their drives home [25]. However, the risk of a traffic collision is not the only way that sleep deprivation puts doctors at risk. Aside from a general increase in risk of infection, decreased leukocytes and enhanced inflammatory pathways [4], sleep deprivation can also have consequences on the metabolism, increasing the risk of obesity and type 2 diabetes [26], as well as contributing to the development of a depressed mood [3]. Furthermore, it has been shown that a constant restriction of sleep over a period of 14 days has a similar effect on neurobehavioral functions as 2 days of total sleep deprivation [27], meaning doctors might not just be slightly sleepy, but dangerously tired. While all these studies link sleep deprivation and decreased performance, as well as poor health, there are few studies that objectively measure and compare the time that medical residents spend sleeping. Further, even fewer studies draw comparisons between the specialties or to the general public. As a result, there is little evidence of how much sleep doctors actually obtain on a regular basis. Consequently, there are no assessments of whether the stereotypes of chronic sleep deprivation are accurate and if the average resident may be experiencing the abovementioned consequences of it. It was the aim of this systematic review to collect and analyze those studies that do objectively measure doctors’, specifically residents’, sleep times and compare them, not only to the general guidelines for sleep time and the national average, but also to contrast the average time slept by residents of different medical specialties. 

In our online search for literature, we could not identify any other reviews assessing the same question. As mentioned above, a potential lack of sleep could affect doctors’ everyday lives and the quality of medical care they can provide. Hence, we regard the definitive assessment of sleep times which this review aims to analyze as important and novel.

## 2. Materials and Methods

The PRISMA (Preferred Reporting Items for Systematic Reviews and Meta-Analyses) guidelines for systematic review and meta-analysis were followed (Citation PRISMA). This review’s protocol was registered with the International Platform of Registered Systematic Review and Meta-Analysis Protocols (INPLASY) on the 15th of November 2022 and was last updated on the 29th of November 2022 (Registration number: INPLASY 202270074). In accordance with the PRISMA guidelines, we aimed to summarize the average sleep times of medical residents and compare and contrast the mean amount of sleep achieved in regard to the respective medical specialties.

A systematic search was conducted via the online archive PubMed and returned 792 results for the combination of the search terms “Sleep” and “Resident” as well as “Registrar”, as to include research from the UK and Australia, where residents may also be referred to as registrars. Figure 1 illustrates the flow chart of the systematic literature search according to PRISMA guidelines. Following by reviewing the titles and excluding those papers that did not refer to medical residents specifically, these results were narrowed down to 348 papers, which were subsequently narrowed further by review of their abstracts, resulting in a total of 41 papers remaining for screening of full-text eligibility. On reading these, a further 11 papers were excluded for various reasons as depicted in Figure 1, leaving a total of 30 papers to be reviewed. To facilitate a comparison and eventually allow us to answer our research questions, these papers all had to include a value in hours of time slept by medical residents. In papers where sleep times for other medical personnel were also assessed, the sleep times of residents had to be stated separately from the sleep times of the other groups to allow for a meaningful statistical comparison.

Statistical analysis was performed by using SPSS Statistics version 27 (International Business Machines Corporation, Armonk, NY, USA). A one-way analysis of variance (ANOVA) test with a post hoc Tukey HSD test, to compare the multiple groups amongst themselves, were performed on a subset of the data collected to create a comparison between the mean sleep times of different medical specialties. Additionally, a Mann–Whitney U test was conducted to assess differences in the data collected by questionnaire and actigraphy. For all analyses, *p*-values < 0.05 were considered statistically significant.

## 3. Results

### 3.1. Distribution of Papers

The papers which were analyzed (Table A1) were from 12 different countries of origin, although they were not evenly distributed amongst these, with 53% of papers stemming from the United States and most other countries contributing one paper, with the exception of Brazil, Saudi Arabia and Canada, which contributed two (Figure 2).

In the literature analyzed, sleep times were always collected by one of three different methods: actigraphy, daily sleep logs or diaries and one-time questionnaires. While other means of measuring sleep time are available, they were not utilized in any papers which matched our inclusion criteria, and thus could not be analyzed in this review. The questionnaires varied between each study, but mostly assessed a range of lifestyle factors and various means of measuring sleepiness, such as the Epworth Sleepiness Scale. Of importance for this review was the fact that they all required residents to give an estimation of their daily average sleep times. The distribution of data collection methods is shown in Table 1. 

Data for 13 different specialties were collected, as well as some data for which the specialty was not specified in the paper, or the results of the study were not split up by specialty (Figure 3). Data from papers which did not separate by specialty were logged under “multiple” and used as a guideline for sleep time of medical residents in general. The most common specialty studied was surgery, for which there were seven sources, closely followed by emergency medicine and pediatrics with five sources each and anesthesiology, internal medicine, psychiatry and urology, which had four sources.

The most common metric for sleep was the mean sleep time, both on and off call, which was assessed in 24 out of 30 papers. Because duties and call systems vary not only by country but also from hospital to hospital, only sleep time off call was analyzed and is represented in Figure 4. The graph shows that the mean sleep time for most specialties is around 6 h, with pediatricians seemingly achieving the most sleep (mean of 7.54 h) and urologists achieving the least sleep (mean of 4.72 h).

### 3.2. Sub-Analysis of Papers from the USA

To better compare mean sleep times between specialties, a sub-analysis of papers stemming from the U.S. was conducted, as a direct comparison between data from different countries does not account for differences in permissible work hours, average workload and culture. Of the 16 papers originating in the USA, 14 assessed mean sleep time, 10 of which also separated their data by specialty. As a result, data were collected for 12 specialties. As described above, data from papers which did not separate by specialty were logged under “multiple”. The mean sleep time varied from 4.88 h for urology to 6.82 h for pediatrics. Apart from urology, surgery and anesthesiology, for which the mean sleep times were 4.88, 5.57 and 5.95 h, respectively, all specialties achieved over 6 h of sleep on average. Notably, none of the specialties reviewed managed to average a sleep time that was over 7 h (Figure 5).

Emergency medicine, psychiatry, family medicine, dermatology, obstetrics/gynecology as well as neurology had to be excluded for this analysis, as only one data point was collected for each, and this is not sufficient for a meaningful statistical analysis. With the exception of the comparison between pediatrics and urology, sleep times between specialties did not differ significantly, with *p*-values ranging between 0.108 and 1. However, when comparing urology to pediatrics, urology residents’ sleep time was significantly shorter than that of their pediatric counterparts (*p* = 0.039).

### 3.3. Comparison of Data Collection Methods

Because the sleep data collected in the papers analyzed were collected using two different methods, a further analysis was conducted to assess whether the form of data acquisition makes a difference to the sleep time recorded. Sleep times were recorded either by a wrist-worn actigraph [44,45,46,47,48,49,50,51,52,53], such as a FitBit, or through a questionnaire [54,55,56,57]. A boxplot (Figure 6) comparing the two methods shows that while the mean sleep times are similar, 6.1 and 6.6 h, respectively, the data collected by actigraph have a greater variance, with a value of 1.33 compared to a variance of 0.32 for data collected by a questionnaire.

However, the results of the conducted Mann–Whitney U test show that the recorded mean sleep times do not differ significantly when comparing the two assessment methods (*p* = 0.759).

## 4. Discussion

While there where barely any significant differences in sleep time between different medical specialties, a common point to all data collected was that the recommended daily sleep amount of 7 h was generally not achieved. Only 5 [35,37,47,49,51] out of the 24 papers which cited a mean sleep time recorded a time over 7 h, and even in these cases the sleep times did not exceed 8 h. Notably, there was only one significant difference in sleep times when specialties were compared to each other, with the average 4.88 h of sleep that resident urologists obtain being significantly shorter than the 6.82 h of sleep the average pediatric resident achieves. A comparison of all other specialties to each other showed no significant differences in sleep time, implying that sleep deprivation among residents is caused more by the general structure and workload of the job and less by the specific tasks and differing routines of individual specialties. 

What this also implies is that it is difficult for a single department or hospital to enact a positive change to the situation if the correct framework, for instance in the form of further duty hour restrictions, is not given. Many countries already have laws in place which limit the maximum weekly work hours permitted. However, the restrictions are often seemingly not tight enough. In the USA, for example, the work hour restriction limits doctors to 80 h per week, meaning that residents may still spend 11 h, almost half their day, at work. Given commute times and responsibilities outside of the workplace, it seems that the framework in place simply does not allow for residents to achieve 8 or more hours of sleep on a regular basis.

Further, considering that on-call sleep times were not analyzed in this review but are on the whole much shorter than the already insufficient off-call sleep times, the overall average of rest achieved will be even smaller. An analysis of the differences between data collected through a questionnaire or by actigraphy showed no significant differences in the sleep times recorded, meaning a comparison of the data is possible, even though these data have been collected in different ways. It should, however, be noted that the sample sizes of those studies that recorded data via actigraphy were considerably smaller. With the exception of one study [51] which managed to recruit 302 residents, participant numbers for actigraphy studies were in the range of 10–39 participants. Comparatively, the sample sizes for studies using questionnaires ranged from 196 to 3604. Seeing as the statistical analyses in this review were only performed on data collected in the U.S., it is unclear as to how far these results apply to other countries. While the papers collected for this review from outside the U.S all cite similarly low sleep times, the number of papers from each country was not sufficient for a direct comparison of data. Regardless, it seems that no matter the country or specialty residents practice in, chronic sleep deprivation is an issue all of them contend with on a daily basis. Considering the consequences of sleep deprivation addressed before, the data imply that most doctors are not given the chance to practice medicine in an environment that is safe for them or their patients, as their attention span and cognitive performance are decreased [3], which invariably leads to a greater number of mistakes and accidents.

As aforementioned, the data concerning a decrease in surgical skills in sleep deprived residents are conflicting. It is also of note that all of the papers citing a decline in surgical skills when sleep deprived measured skill level using various simulated tasks [6,7,8,9,10,11,12,13,14,15,16,17,18,19,20,21,22,24,58,59,60,61,62,63,64,65,66,67,68]. These findings may not be directly transferrable to actual everyday surgeries, as surgeons are rarely operating alone in these situations, so mistakes may be caught or corrected by colleagues. 

Furthermore, surgeons being assessed via simulation invariably know that their skills are being tested and so may be subject to the Hawthorne effect, meaning that their overall performance in the simulation may be better than it would be in the real-world conditions of an OR. 

Therefore, retrospective studies should also be taken into account when assessing the effects of tiredness on surgical skills. 

One retrospective study comparing rates of complications between laparoscopic surgeries performed at night versus during the daytime found a significant increase in intraoperative complications when the procedures were performed between 5 p.m. and 8 a.m. [69]. Furthermore, the risk of complications was higher for surgeries performed during the night, i.e., between 12 a.m. and 8 a.m., compared to surgeries performed early after regular working hours ended, i.e., after 5 p.m. but before 12 a.m. [69]. This suggests that fatigue may have played some role in the increase in complications. However, it is also notable that the surgeries performed during regular working hours had a higher rate of consultants attending, so that a lack of surgical experience must not be discounted as a factor contributing to the higher rate of complications. 

Additionally, the authors state that most of the complications were due to an error on behalf of the anesthesiologist [69]. This may match the findings of Leff et al. [21] that in a sleep deprived state, well-rehearsed technical skills require less concentration than cognitive tasks, meaning that surgeons may be less affected by sleepiness than their colleagues in internal medical specialties. 

On top of this, most studies analyzing the performance of sleep deprivation on surgical performance focus on acute sleep deprivation, i.e., immediately post call [6,7,8,9,10,11,12,13,14,15,16,17,18,19,20,21,22,24,58,59,60,61,62,63,64,65,66,67,68]. 

However, the results of this review imply that most residents are chronically sleep deprived as well, and there are very little data evaluating the effects of chronic sleep deprivation on surgical skills. 

An assessment of these proposed effects of chronic sleep deprivation on medical residents poses an obvious problem: as is suggested by this review, there are barely any medical residents that do not suffer from lack of sleep, meaning that there is no control group of residents who sleep enough and whose performance can be compared to that of their chronically sleep deprived colleagues. 

A partial solution to this issue may be to compare the performance of residents from countries with stricter work hour restrictions compared to those in countries with longer average work hours. However, the current data recording residents sleep times outside of the USA are not sufficient to meaningfully compare it internationally. Moreover, this proposal does not take into account general differences in medical care and training between different countries, which may make it difficult to compare resident performance internationally.

On the other hand, the physical implications of sleep deprivation, including increased risks for infection [4], diabetes and obesity [26], mean that residents are sacrificing their own health in order to treat others. Another important factor to consider is the result of a 2003 study on sleep deprivation [27], which showed that after 14 days of constant sleep restriction to less than 6 h, subjects’ perceived sleepiness plateaued or even declined, although they performed worst in objective cognitive tests. The implication of this is that residents may believe they have “adjusted” to obtaining less sleep, thus not prioritizing achieving enough rest because they do not feel as sleepy, all while their objective cognitive performance will continue to decline. The status quo of the sleep deprived doctor seems to have led to a work environment where a lack of sleep is no longer viewed as a problem to be dealt with but rather seen as an inevitable part of life, which is to be accepted along with the job title. A shift in perspective on this issue is vital, not only to protect patients, who may suffer as a result of unrested doctors, but also for doctors themselves, who put their physical and mental wellbeing on the line when they do not obtain sufficient rest.

The main limitation to this review was imposed by the amount of primary literature studies detailing the average sleep time of medical residents. In our online search, we could only identify 30 such papers. These data were especially lacking in countries outside of the North American continent, making it hard to compare and contrast sleep times between residents of different nationalities. 

Further, it is hard to account for cultural differences and differing workplace structures between the various international publications identified. Hence, a direct comparison of the sleep times between different specialties is only really possible on a country-specific basis. As stated above, this comparison of specialties is currently only possible for the USA, as the data from other countries are not sufficient for a meaningful statistical analysis.

## 5. Conclusions

In summary, an analysis of various papers citing sleep times of residents from a variety of countries, cultures and specialties shows that regardless of these dividing factors, a lack of sleep is a common point which seemingly unifies all medical residents. A second unifying factor seems to be the way that the myriad of negative side effects this chronic sleep deprivation has been largely swept aside, putting both patients at risk of being subjected to mistakes and doctors at risk of poor health and accidents.

In the future, it may be of interest to repeat a review of the literature concerning sleep times of medical residents if further studies collating these data are released. Concerning the abovementioned limitations in comparing residents’ sleep times, a potential, prospective, multi-center study, while surely difficult to accomplish, could provide valuable data which would allow for a more meaningful comparison of sleep times between the various medical specialties. Furthermore, and especially in regard to a possible shift in residents’ wishes for a work–life balance or potential new laws implementing work hour restrictions, it would be interesting to evaluate whether sleep times eventually go up, or whether a shortage of skilled workers and strained medical infrastructure mean residents will continue to be sleep deprived or even whether sleep times may decrease.

## Figures and Tables

**Figure 1 ijerph-20-04180-f001:**
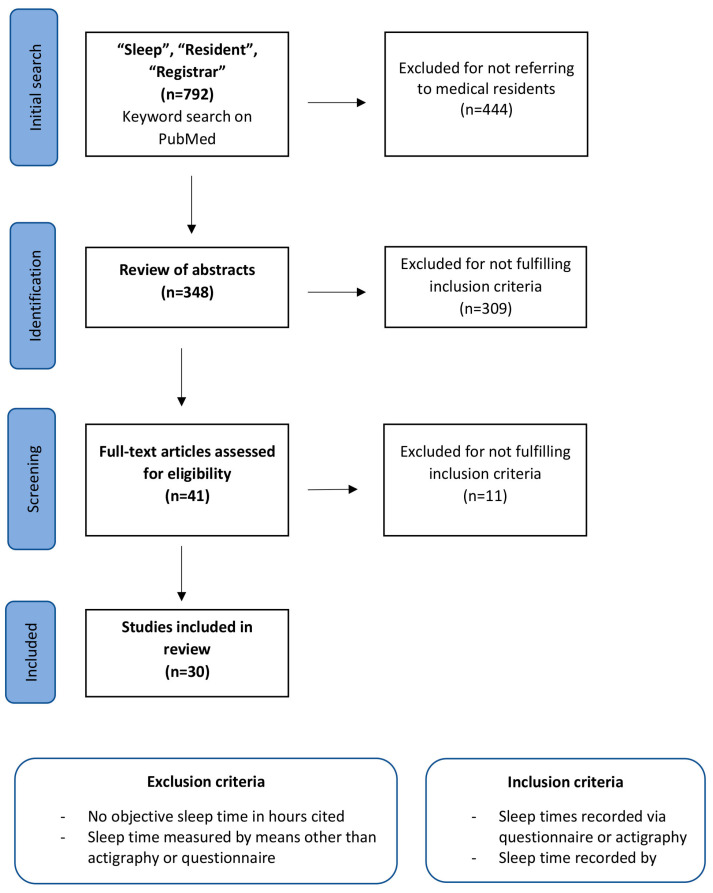
Flow chart depicting the systematic literature search and selection process in chronological order.

**Figure 2 ijerph-20-04180-f002:**
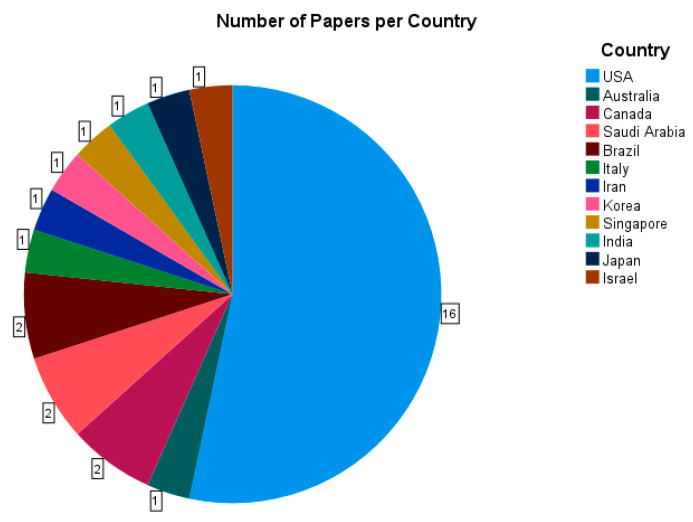
A pie chart analysis of all 30 papers analyzed [28,29,30,31,32,33,34,35,36,37,38,39,40,41,42,43,44,45,46,47,48,49,50,51,52,53,54,55,56,57] by country of their origin.

**Figure 3 ijerph-20-04180-f003:**
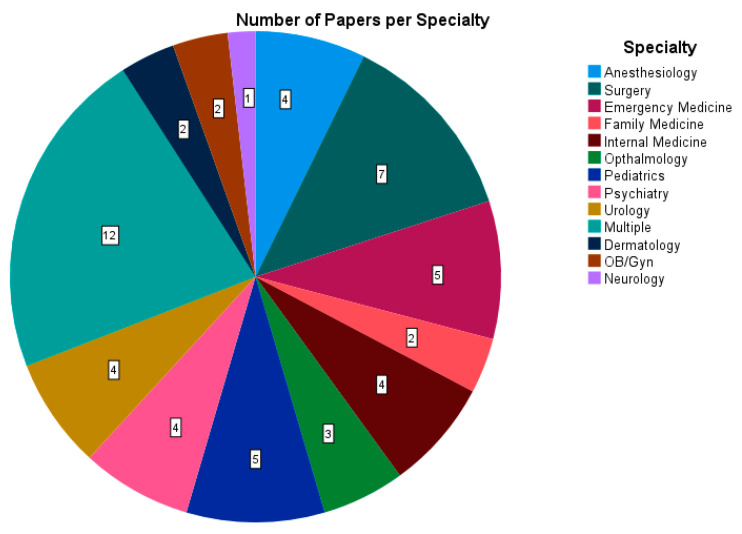
A pie chart analyzing all 30 papers [28,29,30,31,32,33,34,35,36,37,38,39,40,41,42,43,44,45,46,47,48,49,50,51,52,53,54,55,56,57] regarding the number of papers which provided data for each specialty. **Multiple:** the data collected were not divided by specialty.

**Figure 4 ijerph-20-04180-f004:**
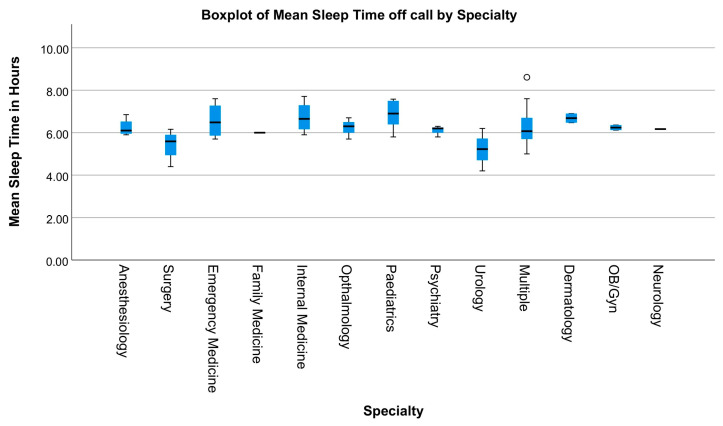
A Box plot of 24 papers [34,35,36,37,38,39,40,41,42,43,44,45,46,47,48,49,50,51,52,53,54,55,56,57] displaying the mean sleep time for each specialty. The circle represents an outlier.

**Figure 5 ijerph-20-04180-f005:**
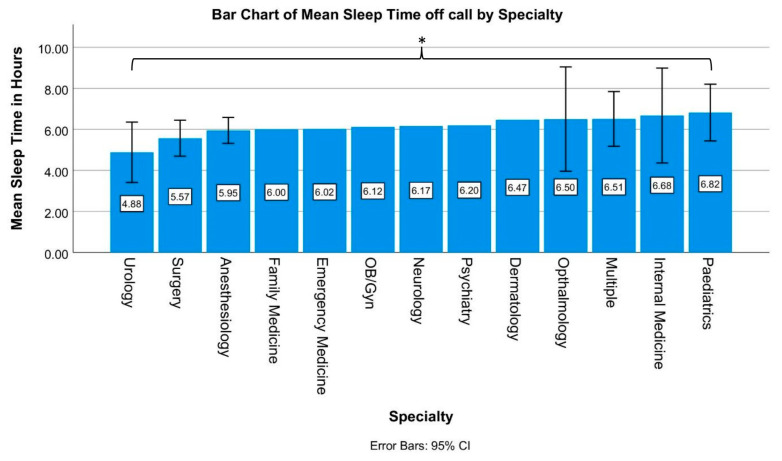
A bar chart analyzing 14 papers [44,45,46,47,48,49,50,51,52,53,54,55,56,57]. The figure displays the mean sleep time of residents practicing in the USA by specialty. **Multiple:** the data collected were not divided by specialty. Significant differences in sleep times are marked with * (*p* < 0.05).

**Figure 6 ijerph-20-04180-f006:**
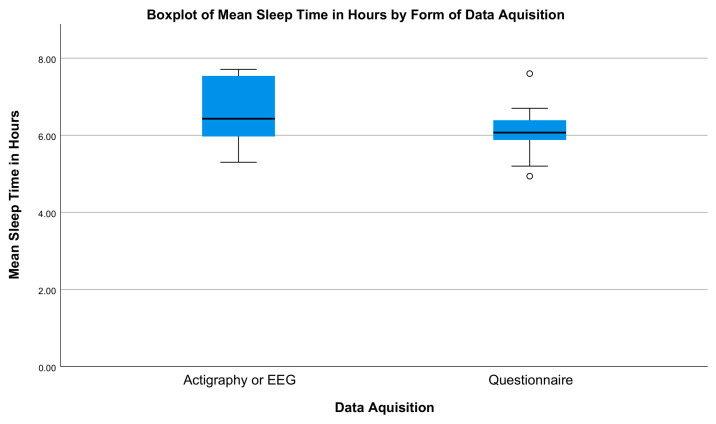
A boxplot using data from 14 papers [44,45,46,47,48,49,50,51,52,53,54,55,56,57] comparing the mean sleep times recorded either by actigraphy or questionnaire. The circles represent outliers.

**Table 1 ijerph-20-04180-t001:** Distribution of data collection method amongst the papers analyzed in this review.

	Actigraphy	Questionnaire	Sleep Log/Sleep Diary	
Papers from USA	8	7	1	
Papers from other countries of origin	6	7	1	
Total	14	14	2	30

## Data Availability

The datasets analyzed during his systematic review are available from the corresponding author in reasonable request.

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
