# Peer review of "Insomnia—A Systematic Review and Comparison of Medical Resident’s Average Off-Call Sleep Times"

_ijerph, 2023, doi:10.3390/ijerph20054180_

Round 1

Reviewer 1 Report

Dear Authors,

this interesting paper is well prepared and described.

However, I have got a few comments:

First of all, insomnia and sleep deprivation is not the same. The topic of the paper is insomnia, in the paper authors mostly describe sleep deprivation. 

Most papers already showed that doctors and residents have sleep deprivation. Statistical analysis did not really show any differences in the results.

It would be interesting to find out factors influencing insomnia or sleep deprivation. Maybe it is not the medical profession that is related to insomnia but other factors. The aim of the study could be analyzing different factors influencing insomnia, not only the differences among specialists.

What questionnaires were used in analyzed papers? What kind of sleep problem did they really evaluate? please add this information.

It is not shown on the figure wat was statistically significant.

What are the limitations of the study?

Thank you for the possibility to read this paper.

Author Response

We would like to thank the reviewers for their careful consideration and review of our manuscript and their constructive comments, which have allowed us to improve the quality of our review. Please find our point-by-point response below:

Reviewer 1:

  1. First of all, insomnia and sleep deprivation is not the same. The topic of the paper is insomnia, in the paper authors mostly describe sleep deprivation.

Most papers already showed that doctors and residents have sleep deprivation. Statistical analysis did not really show any differences in the results.

It would be interesting to find out factors influencing insomnia or sleep deprivation. Maybe it is not the medical profession that is related to insomnia but other factors. The aim of the study could be analyzing different factors influencing insomnia, not only the differences among specialists.

We apologize for any confusion stemming from the title of our manuscript. While we agree that the further questions posed by the reviewer about insomnia and its varying factors are of interest, our aim was not to propose answers for why doctors are not sleeping enough, but rather to asses whether this truly was the case. By using the word “insomnia”, we merely aimed to make out title bolder.

  1. What questionnaires were used in analyzed papers? What kind of sleep problem did they really evaluate? please add this information.

Thank you for addressing the need for clarification. We have further elaborated on the type of questionnaires used in the results section (page 5, line 165-169).

  1. It is not shown on the figure wat was statistically significant.

Figure 5 (page 7) has been amended to show significance between sleep-time differences. As the other figures showed no significant differences between data, a signifier could not be added.

  1. What are the limitations of the study?

Thank you for pointing out this missing section. A limitations section has been added to the discussion (page 10, line 339 – 348).

Reviewer 2 Report

This work showed an interesting investigation for analyzing the medical residents average sleep times. However, I have a number of further comments to improve the manuscript:

Introduction:

·      Line 48 - 63 discuss the reduced performance of surgeons when sleep deprived. It is better to make this in one/ two paragraph. Also, line 64 – 73.

·      However, why does the author only focus on surgeons' problems? As we know, there are so many types of medical residents.  This needs to be clarified in the manuscript based on the literature review.

·      The novelty or state-of-the-art of this manuscript should be described.

Materials and methods:

·      line 109 and 110 should be in the same paragraph.

·      Please describe what data criteria are needed for this research, and why is that so?

Results

·      Table 1: Please revise the table position

·      How authors have selected only 3 different sleep methods (Actigraphy, daily sleep logs or diaries and one-time questionnaires) since there are other sleep method, such as polysomnography and ballistocardiography. It should be commented on in the manuscript.

Discussion

·    Basic sections have been covered, limitations section can be added.

Conclusion

·    Basic sections have been covered, future section can be added.

Author Response

Point by Point Response

We would like to thank the reviewers for their careful consideration and review of our manuscript and their constructive comments, which have allowed us to improve the quality of our review. Please find our point-by-point response below:

Reviewer 2:

  1. Line 48 - 63 discuss the reduced performance of surgeons when sleep deprived. It is better to make this in one/ two paragraph. Also, line 64 – 73.

We have adjusted the formatting of the paragraphs accordingly. (See page 2, line 49-63 and line 64-73)

  1. However, why does the author only focus on surgeons' problems? As we know, there are so many types of medical residents.  This needs to be clarified in the manuscript based on the literature review.

Thank you for addressing this point. As the authors of the manuscript are form a surgical department, the emphasis on surgeon’s problems with sleep deprivation was of special interest. We have elaborated on page 2, line 87-93.

  1. The novelty or state-of-the-art of this manuscript should be described.

A clarification of the novelty of the manuscript has been added to the introduction (page 3, line 117-121).

  1. line 109 and 110 should be in the same paragraph.

We have adjusted the formatting accordingly (page 3, line 123-124)

  1. Please describe what data criteria are needed for this research, and why is that so?

A clarification on the data criteria required has been added to the methods section (page 3, line 140-144)

  1. Table 1: Please revise the table position

We have adjusted the table position (page 5).

  1. How authors have selected only 3 different sleep methods (Actigraphy, daily sleep logs or diaries and one-time questionnaires) since there are other sleep method, such as polysomnography and ballistocardiography. It should be commented on in the manuscript.

Thanks for pointing out the need for clarification. Within the literature that fit our inclusion criteria, these 3 methods of sleep time measurement happened to be the only ones utilised, so that, even though other methods are available, they could not be analysed in our manuscript. We have revised the line to clarify (page 5, line 164-172).

  1. Basic sections have been covered, limitations section can be added

Thank you for pointing out the missing section. A limitations section has been added (page 10, line 339-348)

  1. Basic sections have been covered, future section can be added

Again, we thank you for pointing this out, we have added a future section (page 10, line 356-365)

Round 2

Reviewer 2 Report

The authors well revised the manuscript taking into account all review's comments.